# Rethinking Audio-Visual Adversarial Vulnerability from Temporal and Modality Perspectives

[*]**Zeliang Zhang**[1], [*]**Susan Liang**[1], **Daiki Shimada**[1,2], **Chenliang Xu**[1]
[1]University of Rochester      [2]Sony Group Corporation
{zeliang.zhang, susan.liang, chenliang.xu}@rochester.edu
Daiki.Shimada@sony.com

## Abstract

While audio-visual learning equips models with a richer understanding of the real world by leveraging multiple sensory modalities, this integration also introduces new vulnerabilities to adversarial attacks. In this paper, we present a comprehensive study of the adversarial robustness of audio-visual models, considering both temporal and modality-specific vulnerabilities. We propose two powerful adversarial attacks: 1) a temporal invariance attack that exploits the inherent temporal redundancy across consecutive time segments and 2) a modality misalignment attack that introduces incongruence between the audio and visual modalities. These attacks are designed to thoroughly assess the robustness of audio-visual models against diverse threats. Furthermore, to defend against such attacks, we introduce a novel audio-visual adversarial training framework. This framework addresses key challenges in vanilla adversarial training by incorporating efficient adversarial perturbation crafting tailored to multi-modal data and an adversarial curriculum strategy. Extensive experiments in the Kinetics-Sounds dataset demonstrate that our proposed temporal and modality-based attacks in degrading model performance can achieve state-of-the-art performance, while our adversarial training defense largely improves the adversarial robustness as well as the adversarial training efficiency.

## 1 Introduction

Audio-visual models, capable of integrating both auditory and visual information, have gained significant traction in recent years due to their ability to create a comprehensive understanding of the surrounding world (Li et al., 2022a; Song et al., 2023; 2024a;b). These models have demonstrated remarkable success in a wide range of applications, including multimedia analysis (Dimoulas, 2016), human-computer interaction (Zhen et al., 2023), and autonomous systems (Guo et al., 2023). However, a critical challenge lies in their susceptibility to adversarial attacks (Zhang et al., 2024b;a; Yao et al., 2024). These attacks can craft imperceptible perturbations to the input data, causing audio-visual models to make erroneous predictions (Wang et al., 2022) or interpretations (Han et al., 2023). Such errors can have disastrous consequences, especially in safety-critical domains like auto-driving (Kloukiniotis et al., 2022) and identity verification (Zhang et al., 2021).

While prior work has investigated the adversarial robustness of audio-visual models (Tian & Xu, 2021; Yang et al., 2021; Li et al., 2022b), they primarily rely on general adversarial attack methods, such as FGSM (Goodfellow et al., 2015) and I-FGSM (Kurakin et al., 2017), originally designed for the single-modality data. These methods are simply adapted to the audio-visual domain without fully capitalizing on its unique characteristics. A key limitation of such approaches lies in their inability to consider the inherent properties of audio-visual data. Unlike single modality, audio-visual data possesses two crucial aspects: temporal consistency and intermodal correlation. For instance, in a video of a dog barking, we *see* and *hear* the barking event *unfold over time*, not just in a single frame. These properties play a vital role in human perception of the real world (Sun et al., 2022;

---

[*]Equal contribution. Listed order is random.

Yang et al., 2023a). However, current attack methods fail to exploit them, potentially limiting their effectiveness. Conversely, by leveraging these characteristics, we can craft more potent attacks and develop improved robust learning strategies specifically tailored for audio-visual models.

In this work, we rethink the adversarial vulnerability of audio-visual models through the lenses of temporal and modality perspectives. We begin with an empirical analysis to assess the vulnerability of existing models. Our case study experiments reveal several key findings, including the presence of adversarial transferability within the audio-visual domain, and the significant impact of temporal consistency and modality correlations on model robustness. Leveraging these insights, we propose two novel adversarial attacks tailored to the unique properties of multi-modal data: 1) the temporal invariance attack, which targets robust and temporally consistent audio-visual features by introducing inconsistencies across consecutive frames, and 2) the modality misalignment attack, which crafts adversarial examples by inducing incongruencies between the audio and visual streams.

To mitigate the vulnerabilities exposed by these dedicated attacks, we propose a novel audio-visual adversarial training framework that serves as a robust defense mechanism. Our framework addresses critical challenges in robust multi-modal learning by incorporating efficient adversarial perturbation crafting techniques along with an adversarial curriculum training strategy. The proposed defense aims to significantly improve the robustness of audio-visual models against adversarial attacks with minimal impact on training efficiency.

Our contributions can be summarized as follows:

1. We first identify the existence of adversarial transferability in audio-visual learning, and introduce two powerful adversarial attacks, namely the Temporal Invariance-based Attack (TIA) and the Modality Misalignment-based Attack (MMA), to evaluate the adversarial robustness of audio-visual models comprehensively.

2. We propose efficient adversarial perturbation crafting and adversarial curriculum training aimed at enhancing both the robustness and efficiency of audio-visual models.

3. We validate the effectiveness of both our proposed attacks and defense mechanisms through extensive experiments conducted on the widely-used Kinetics-Sounds dataset.

## 2 RELATED WORK

### 2.1 AUDIO-VISUAL LEARNING

The field of audio-visual learning encompasses a wide range of tasks, including audio-visual event recognition (Brousmiche et al., 2019; Xia & Zhao, 2022; Brousmiche et al., 2022), separation (Wu et al., 2019a; Majumder et al., 2021; Majumder & Grauman, 2022; Huang et al., 2023a), localization (Wu et al., 2019b; 2021; Huang et al., 2023b), correspondence learning (Min et al., 2020; Zhu et al., 2021; Morgado et al., 2021), representation learning (Zhou et al., 2019; Cheng et al., 2020; Rahman et al., 2021), and cross-modal generation (Chen et al., 2017; Hao et al., 2018; Sung-Bin et al., 2023; Liang et al., 2023a; 2024; Huang et al., 2024; Liang et al., 2023b). Among these, audio-visual event recognition stands out as a fundamental task (Gao et al., 2024) that has attracted significant research attention, particularly regarding robustness and security issues (Yang et al., 2023b).

Deep learning models employed in audio-visual event recognition typically comprise three main components: visual encoder, audio encoder, and fusion layer. Although prior research has extensively focused on optimizing these components to enhance task performance, there has been limited consideration of their implications for security and robustness.

In this work, we delve into the individual components of these models and examine their respective impacts on robustness, shedding light on crucial but often overlooked aspects.

### 2.2 ADVERSARIAL ATTACK & DEFENSE

Research efforts in adversarial robustness for audio-visual models have been relatively limited. Tian *et al.* (Tian & Xu, 2021) were among the first to explore the potential of audio-visual integration in enhancing robustness against multi-modal attacks. Yang *et al.*(Yang et al., 2021) proposed an adversarially robust audio-visual fusion layer to defend against single-source adversarial attacks.

Li *et al.* (Li et al., 2022b) introduced a novel mix-up strategy in the audio-visual fusion layer to improve the robustness of audio-visual models. Yang *et al.* (Yang et al., 2023b) proposed a certified robust training method to boost the multi-modal robustness. However, they primarily focused on adapting single-modality adversarial attacks to audio-visual scenes. There remains a critical need for powerful audio-visual adversarial attacks that can serve as benchmarks for evaluating the adversarial robustness of audio-visual methods and the effectiveness of robust training techniques.

In this work, we address the gap by designing an effective audio-visual adversarial attack method that facilitates a more comprehensive assessment of model robustness. We further propose an efficient defense technique to enhance the robustness of audio-visual models against adversarial attacks.

## 3 EMPIRICAL ROBUSTNESS ANALYSIS OF AUDIO-VISUAL MODELS

**Notations.** Given an audio clip $x_a$ and video frames $x_v$, we use an audio network $f_a(x_a; \theta_a)$ to extract audio features, a visual network $f_v(x_v; \theta_v)$ to extract visual features and a fusion network $f_u$ for modality integration. We denote the complete audio-visual network with $F(x_v, x_a; \theta) := f_u(f_v(x_v; \theta_v), f_a(x_a; \theta_a); \theta_u)$, where $\theta = (\theta_v, \theta_a, \theta_u)$ are the overall network parameters.

Temporal consistency and modality correlation are two fundamental characteristics of audio-visual learning. On the one hand, *these characteristics provide the robustness and generalization capabilities to audio-visual models*. The temporal consistency in audio-visual data reinforces learning across consecutive frames, while the cross-modal correlations between auditory and visual signals offer mutually complementary information. These properties enable audio-visual models to learn robust and reliable representations. However, *the temporal and cross-modal dependency also paradoxically create new vulnerabilities*. Unlike conventional attack methods, audio-visual adversarial attacks can exploit these relationships to cause inconsistencies within the model, leading to errors. This duality underscores the critical need to understand and address adversarial robustness in audio-visual models from both temporal and modality perspectives.

**Corruption Robustness**. Here, we provide experiments to support our arguments. We train an audio-visual model on the Kinetics-Sounds dataset (Arandjelovic & Zisserman, 2017), which takes the VGG as the vision encoder, the AlexNet as the audio encoder, and the sum operation as the fusion operation followed by the decision layer. We randomly mask a ratio of $\rho$ of the audio-visual data along the temporal dimension, where $0\% < \rho < 30\%$, and evaluate the model performance on the masked audio-visual data. For a comprehensive understanding of how different modalities affect the audio-visual model's decision, we set up three groups, namely perturbing the video only (V×), perturbing the audio only (A×), and perturbing both the audio and visual synchronously (A×⊕V×) and asynchronously (A×⊗V×).

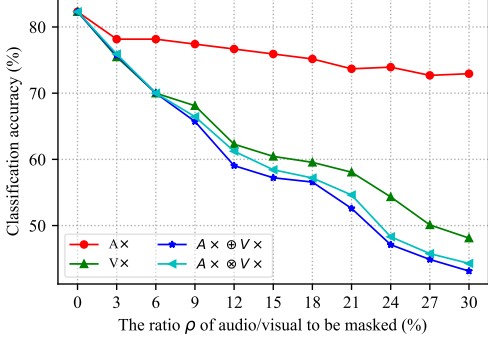

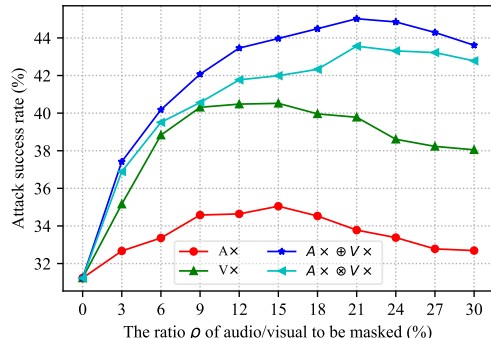

Figure 1: Classification accuracy of the model when masking the audio and visual data with a ratio of $\rho$.

Figure 2: Average black-box attack success rate (%) against 7 black-box models.

We have three observations from the results shown in fig. 1. First, the audio-visual data is relatively redundant in temporal information. By masking the audio-visual data with certain ranges, *e.g.*, $\rho < 20\%$, the model remains at least $50\%$ classification success rate, indicating robustness against

the temporal perturbation. Second, the model heavily relies on the visual modality to make the decision, leaving the audio modality less attention. With the same ratio $\rho$ to be masked, the degradation of model performance caused by perturbing the visual data is significantly greater than perturbing the audio data only with a clear margin of $12.3\%$. Third, the correlations between different modalities increase the performance of audio-visual learning. Compared to asynchronous perturbation of audio and visual data, synchronous perturbation causes relatively larger performance degradation, indicating the complementary of two modalities in helping the model make decisions.

**Adversarial Robustness**. For a given audio-visual input $(x_a, x_v)$ with ground-truth $y$, the goal of the adversarial attack is crafting perturbation $\delta_a$ (audio) and $\delta_v$ (visual) to deceive $F$ into making a wrong prediction. This process is formulated as follows,

$$
\begin{aligned}
\max \mathcal{L}(F(x_v + \delta_v, x_a + \delta_a; \theta), y) \\
s.t. \ \|\delta_v\|_p \leq \epsilon_v, \ \ \|\delta_a\|_p \leq \epsilon_a,
\end{aligned}
\tag{1}
$$

where $\mathcal{L}$ is an arbitrary loss function, $\|\|_p$ is the $p$-norm, and $\epsilon_v$ and $\epsilon_a$ are the adversarial perturbation budgets for visual and audio modalities, respectively.

For a further study of the impact of temporal consistency and modality correlation on the adversarial robustness, we additionally train various audio-visual models on the Kinetics-Sounds dataset to evaluate the adversarial attack performance. We, respectively, use the VGG and ResNet as the vision backbone, the AlexNet and ResNet as the audio backbone, and the sum and concat as the fusion layer, in a total of $8$ models. We set the model with VGG as the vision backbone, AlexNet as the audio backbone, and concat as the fusion layer, as the surrogate model to generate adversarial examples by FGSM (Goodfellow et al., 2015) under the white-box setting, which is up to $78.3\%$ attack success rate. Then, we set the other 7 models as black-box models to further evaluate adversarial transferability.

As shown in fig. 2, we can find that adversarial transferability between different models also exists in the audio-visual data. Without any precomputed masking operation, the generated audio-visual adversarial examples have an average attack success rate of $31.7\%$ against the selected black-box models. By masking different modalities along the temporal dimension adequately with ratio $\rho$ to generate multiple copies for gradient calculation, the adversarial transferability can be boosted. Also, a high setting of $\rho$ causes a loss of information, hindering the quality of generated adversarial examples and degrading the adversarial transferability. Specifically, it can bring up to an improvement of $3.8\%$ by only masking the audio, $9.3\%$ by only masking the video, and $13.8\%$ by masking both the audio and visual modality. These results uncover that temporal redundancy in audio and visual data can be naturally used to boost adversarial transferability. Besides, the compensation function from the modality correlation mitigates the impact of adversarial transferability with up to $1.5\%$, comparing the synchronous and asynchronous perturbation.

> **Takeaways of our empirical study**
>
> (1) The adversarial transferability is also exhibited in audio-visual learning, posing security problems in applications.
> (2) Temporal consistency and redundancy bring robustness against corruption but also remain potentially leveraged to improve adversarial transferability.
> (3) The inter-modal correlation compensates against temporal corruptions and alleviates the influence of adversarial perturbation.

## 4 AUDIO-VISUAL ADVERSARIAL ATTACK

Motivated by previous empirical robustness analysis on temporal consistency and modality correlation, we propose two powerful audio-visual adversarial attacks, namely the temporal invariance-based attack and the modality misalignment-based attack.

### 4.1 TEMPORAL INVARIANCE-BASED ATTACK

Audio-visual data comprises temporally invariant features and time-varying information within each frame. Our goal is to craft adversarial perturbations that target these invariant features, thereby

fostering the transferability of adversarial instances. We achieve this by introducing a temporal regularization term that steers perturbations toward the most significant video features. With this novel regularization, we ensure that the adversarial perturbations are consistent and coherent across different frames.

Specifically, we calculate the variation of extracted features along the temporal dimension as a structure-unrelated statistic of feature consistency. By minimizing this variation for both audio and visual features, we encourage perturbations to focus on temporally invariant characteristics.

Additionally, leveraging the inherent temporal dependency, we can diversify video inputs to encourage the model to learn robust, invariant features. In practice, we apply different input transformations on temporal inputs, including scaling, masking, blurring, and mix-up on audio or visual modalities, independently or in parallel, synchronously or asynchronously.

This temporal regularization can be expressed as

$$
\begin{aligned}
\mathcal{L}_R = \mathrm{Var}&\left[\left\{\mathbb{E}\left(f_a\left(\mathcal{T}_a\left(x_a + \delta_a\right); \theta_a\right)(t)\right)\right\}_{t=1}^{T}\right] \\
&+ \mathrm{Var}\left[\left\{\mathbb{E}\left(f_v\left(\mathcal{T}_v\left(x_v + \delta_v\right); \theta_v\right)(t)\right)\right\}_{t=1}^{T}\right],
\end{aligned}
\tag{2}
$$

where we denote $f_a(x_a + \delta_a; \theta_a)(t)$ and $f_v(x_v + \delta_v; \theta_v)(t)$ as the audio and visual features extracted at the $t$-th frame by the audio and visual networks, $\mathcal{T}_a$ and $\mathcal{T}_v$ as input transformation methods for audio and visual modalities, respectively. We consider the mean value of the feature at each time step and compute the variance along the temporal axis.

## 4.2 MODALITY MISALIGNMENT-BASED ATTACK

From the empirical study about the modality correlation in section 3, especially the comparison between the synchronous and asynchronous perturbation, we notice that aligning semantic changes across modalities can mitigate the influence of adversarial perturbations and diminish the adversarial transferability. Inspired by this, we propose a novel attack strategy that disrupts the strong semantic correlation between audio and visual modalities. We hypothesize that **lower semantic correlation leads to higher adversarial audio-visual transferability**.

We conduct experiments to support this hypothesis. To quantify semantic information, we use feature vectors from both modalities and assess their alignment with cosine similarity. Following the experimental setting in section 3 (without masking operation), we compute cosine similarities between feature vectors of adversarial examples generated by FGSM (Goodfellow et al., 2015), I-FGSM (Kurakin et al., 2017), MI-FGSM (Dong et al., 2018), and NI-FGSM (Lin et al., 2019), which exhibit progressively stronger attack performance under the white-box setting.

From the results shown in fig. 3, we can see that as the attack success rate against black-box models (*i.e.*, adversarial transferability) increases, the cosine similarity between audio and visual feature vectors decreases. This correlation reinforces our claim that disrupting semantic alignment between modalities enhances transferability.

Thus, to enhance the transferability of adversarial attacks, we propose an approach that aims to misalign the semantic correspondence between audio and visual features. Specifically, during each iteration of the adversarial attack, we minimize the feature similarity between the modalities, ensuring that the perturbations disrupt the semantic alignment at the modality level,

$$
\mathcal{L}_M = \frac{f_a(x_a + \delta_a; \theta_a) \cdot f_v(x_v + \delta_v; \theta_v)}{\|f_a(x_a + \delta_a; \theta_a) \cdot f_v(x_v + \delta_v; \theta_v)\|_2},
\tag{3}
$$

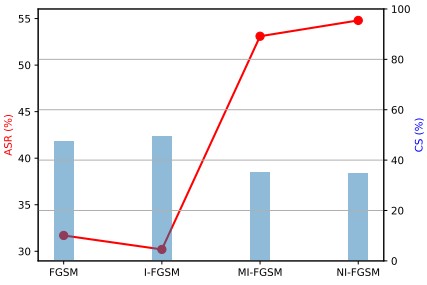

Figure 3: The average black-box attack success rate (A.S.R.) and cosine similarity (C.S.) of the audio-visual adversarial examples generated by different attacks.

where $f_a(x_a + \delta_a; \theta_a)$ and $f_v(x_v + \delta_v; \theta_v)$ are the feature vectors encoded by the audio and visual backbones, respectively.

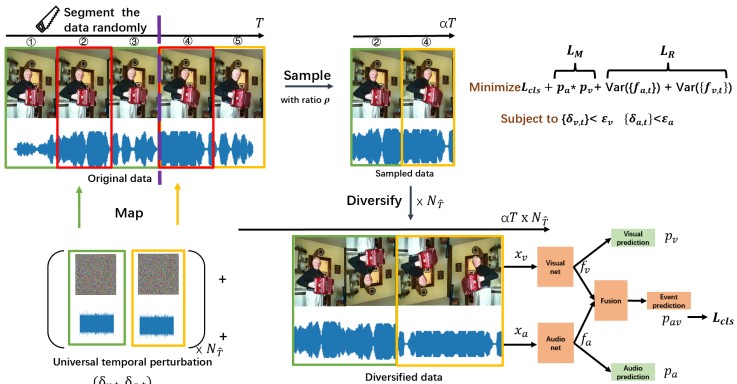

Figure 4: Overview of the adversarial perturbation crafting in the adversarial training process. Given an audio-visual data, we randomly segment it into different parts (green and yellow) and sample frames from each of the segments with ratio $\alpha$ (red). Then, we diversify each sampled frame by $N_{\hat{T}}$ copies and employ TIA and MMA to craft the adversarial perturbation. Finally, we map the generated adversarial perturbation to corresponding segments, creating adversarial examples for adversarial training.

## 4.3 ATTACK INTEGRATION

The temporal invariance-based and modality misalignment-based attacks can be integrated together with the classification loss to achieve strong adversarial audiovisual attacks. At each iteration, we optimize the following loss function to conduct the attack,

$$
\begin{aligned}
\max \ &\mathcal{L}_{cls}(F(\mathcal{T}(x_v + \delta_v, x_a + \delta_a); \theta), y) \\
&- \lambda_1 \mathcal{L}_R(F(\mathcal{T}(x_v + \delta_v, x_a + \delta_a); \theta)) \\
&- \lambda_2 \mathcal{L}_M(F(\mathcal{T}(x_v + \delta_v, x_a + \delta_a); \theta)) \\
s.t. \ &\|\delta_v\|_p \leq \epsilon_v, \ \ \|\delta_a\|_p \leq \epsilon_a,
\end{aligned}
\tag{4}
$$

where $\mathcal{L}_{cls}$ is the classification loss function, $\mathcal{T}$ is the input transformation on audio-visual data, and $\lambda_1$ and $\lambda_2$ are two coefficients to balance losses.

## 5 AUDIO-VISUAL ADVERSARIAL TRAINING

Having established the powerful audio-visual adversarial attacks, a question naturally arises: *How can we defend these attacks efficiently?* In this section, we start with the preliminary adversarial training method, followed by an in-depth analysis of adversarial perturbations in audio-visual data, and propose several strategies to fortify audio-visual models against such attacks.

### 5.1 PRELIMINARY METHOD

Adversarial training is one of the most powerful robust training paradigms to defend adversarial examples, which trains the model on adversarial examples and can be formalized in the audio-visual context as follows,

$$
\min_{\theta} \max_{\delta_v, \delta_a} \mathcal{L}_{cls}(F(x_v + \delta_v, x_a + \delta_a), \theta),
\tag{5}
$$

which optimizes the model parameters to minimize the upper bound of the loss function.

It is challenging to directly optimize the min-max problem in eq. (5). In practice, it is solved by training models on adversarial examples, in which adversarial examples are used to compute the values of the inner maximum loss function (Madry et al., 2018).

### 5.2 DISCUSSION ON AUDIO-VISUAL PERTURBATION

Audio-visual adversarial training presents a unique challenge compared to its uni-modal counterpart (*e.g.,* perturbating a single image). Here, generating adversarial perturbations across multiple time

steps in both audio and visual data leads to significant computational overhead. This hinders the use of strong attacks and advanced mitigation techniques during training. Thus, the key to solving the challenge is to reduce the computational cost for audio-visual adversarial example crafting.

As shown in section 3, random masking of audio-visual data slightly affects model performance yet boosts adversarial transferability. Additionally, recent studies (Kim et al., 2023) demonstrate that the universal adversarial perturbation of a single image model can be used to fool the video model. These findings indicate that an effective strategy should reduce the computational overhead of adversarial training by generating temporal universal adversarial perturbations instead of frame-specific perturbations. However, strong attacks are crucial for effective adversarial training. To address this, we propose generating universal adversarial perturbations for continuous local regions in the video and audio data, rather than a single perturbation for all frames. We also employ a bag of tricks to amplify the impact of these generated perturbations on the model.

## 5.3 OUR APPROACH

**Efficient adversarial perturbation crafting**. Since audio-visual data exhibits strong temporal correlation, adversarial perturbations crafted from a subset of time samples can be propagated to the remaining samples as well.

To exploit this property and achieve efficient perturbation crafting, we propose the following approach: We first divide the entire audio-visual data into smaller segments. Within each segment, we randomly sample a portion of audio and visual frames with a selection ratio of $\rho$. We then generate universal adversarial perturbations upon the selected frames and propagate them to the remaining. The optimization of perturbations $\delta_v$ and $\delta_a$ can be expressed as follows,

$$\delta_v, \delta_a = \arg\max_{\hat{\delta}_v, \hat{\delta}_a} \mathbb{E}_{(\hat{x}_v, \hat{x}_a) \sim (x_v, x_a)} [\mathcal{L}(\hat{x}_v + \hat{\delta}_v, \hat{x}_a + \hat{\delta}_a, y)], \tag{6}$$

where the $(\hat{x}_v, \hat{x}_a)$ is sampled from $(x_v, x_a)$, $\mathcal{L}$ is the loss we proposed in eq. (4) which encounters both the temporal invariance-based and modality misalignment-based attacks. The adversarial perturbations generated $\delta_v$ and $\delta_a$ are shared with neighboring frames within the same segment. This design allows us to generate adversarial examples efficiently.

**Adversarial curriculum training**. Previous studies (Yu et al., 2022; Kim et al., 2021; Rice et al., 2020) have identified that adversarial training easily gets overfitted to certain attack methods and settings, leaving the model vulnerable to others. To address this issue and improve the generalization ability, we propose a randomized adversarial curriculum learning to optimize eq. (5). Concretely, our proposed randomized adversarial curriculum learning approach incorporates two strategies:

- **Data-level strategy**. Temporal redundancy can be leveraged to control the impact of adversarial examples crafted in adversarial training, leaving the potential to alleviate overfitting. We propose to randomly sample $\rho_x$ of the audio-visual input along the temporal dimension, where $0 < \rho_x < 1$. We cyclically vary the value of $\rho_x$ to generate adversarial examples for curriculum learning.

- **Model-level strategy**. The over-parameterization of models can be exploited in adversarial training to boost the models' generalization. We randomly drop out the neurons of the fusion layer with a ratio of $\rho_f$, where $0 < \rho_f < 1$ and the dropout ratio $\rho_f$ is synchronized with $\rho_x$.

The curriculum learning approach hinges on a cyclical variation in the data-level masking ratio $\rho_x$ and the model-level dropout ratio $\rho_f$. This cyclic design offers an advantage: it gradually increases the training difficulty. During the initial stages of the cycle, both sampling and dropout ratios are lower and the number of steps to generate attacks is reduced. This translates to a simpler training task with fewer adversarial perturbations applied to the data. As both ratios and the number of steps gradually increase, complex adversarial examples are created, resulting in a more difficult learning task. With this cyclic design, the models are trained on diverse perturbations, thus fully exploring the loss landscape for the searching minimum and improving the robustness against attacks.

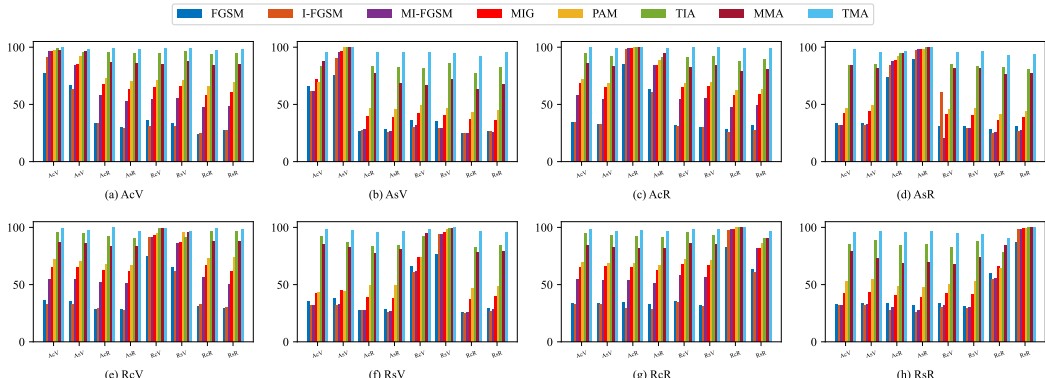

Figure 5: Attack success rates (%) of eight deep models, where the adversarial examples are generated on the white-box surrogate model and attack all models (one white-box model and seven black-box models). TIA, MMA, and TMA are our proposed attack methods.

# 6 PERFORMANCE EVALUATION

## 6.1 EXPERIMENTAL SETUP

**Datasets.** We use the Kinetics-Sounds (Arandjelovic & Zisserman, 2017) for evaluation, which contains $1,551,610$ second video clips in 27 human action categories. For model training, we split the dataset into $7 : 2 : 1$ for training, validation, and testing. We also conduct experiments on MIT-MUSIC (Zhao et al., 2018b) for further verification, which is provided in the appendix.

**Models.** The audio-visual model comprises three modules: the visual backbone, the audio backbone, and the audio-visual fusion network. For the audio backbone, we select VGG and ResNet as candidates. For the visual backbone, we select AlexNet and ResNet as candidates. We can study the impact of model capacity on the adversarial transferability by this design. Following the previous work on audio-visual adversarial robustness (Tian & Xu, 2021), we selected the sum and concat operation as candidates for the fusion layer. There are $2 \times 2 \times 2 = 8$ models in total. For simplicity, we use the format of "{ visual backbone }-{ fusion layer }-{ audio backbone }" to represent the audio-visual models, where the initials indicate each backbone and layer. We denote AlexNet as A, VGG as V, ResNet as R, sum operation as s, and concat operation as c. For example, ResNet as the visual backbone, the audio-visual model with AlexNet as the audio backbone, and the sum operation as the fusion layer can be represented by "RsA".

**Baselines.** For attack methods, we use FGSM (Goodfellow et al., 2015), I-FGSM (Kurakin et al., 2017), MI-FGSM (Dong et al., 2018), MIG (Ma et al., 2023), and PAM (Zhang et al., 2023) as baselines. We compare the baseline methods with our proposed approaches, which encompass the temporal invariance-based attack (TIA), the modality misalignment attack (MMA), and the integration of TIA and MAA attacks, namely the temporal and modality-based attack (TMA). For defense methods, we use the vanilla adversarial training (AT) (Madry et al., 2018), discriminative and compact feature learning (DCFL) (Tian & Xu, 2021), single-source defensive fusion (SSDF) (Yang et al., 2021), the mix-up strategy for adversarial defense (Mixup) (Li et al., 2022b), and the certified robust multi-modal training method (CRMT-AT) (Yang et al., 2023b).

## 6.2 SINGLE MODEL ATTACK

To validate the effectiveness of our proposed methods, we first compare ours with five popular attacks selected by previous audio-visual robustness research, namely FGSM, I-FGSM, MI-FGSM, MIG, and PAM. Among these methods, FGSM and I-FGSM are initially designed for white-box attacks, while others are designed for transferable adversarial attacks in the image domain. We generate adversarial examples on a single model and test them on the other models. The attack success rates, *i.e.*, and the misclassification rates of the victim model in the adversarial examples crafted are summarized in fig. 5.

The figure shows that audio-visual adversarial examples crafted on white-box models can partially deceive black-box models, confirming adversarial transferability in audio-visual learning. However, unlike in the image domain, momentum doesn't always improve performance. For instance, when using VGG as the audio backbone and AlexNet with a sum fusion layer, FGSM achieves a $31.5\%$ success rate, while MI-FGSM decreases performance by $2.3\%$, indicating momentum's negative impact on audio-visual transferability. This issue is common with models using sum fusion layers. In contrast, models with concat fusion layers show better black-box performance ($+3.6\%$). The results also suggest that improving input diversity with PAM helps momentum perform better over sum fusion layers. Our proposed methods (TIA, MMA, and TMA) consistently outperform others, with TIA exceeding PAM by $15.7\%$ and further improving by $5.2\%$ on average. Combining TIA and MMA achieves a $95.2\%$ success rate across all eight models.

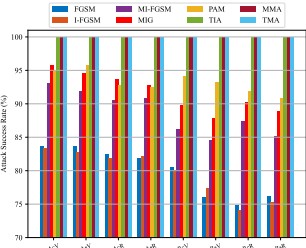

Figure 6: Attack success rates (%) of each of 8 deep models under the ensemble setting. TIA, MMA, and TMA are our proposed attack methods.

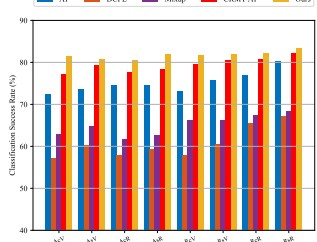

Figure 7: Attack success rates (%) of each of 8 deep models on the adversarial examples crafted under the white-box setting with our proposed TMA method.

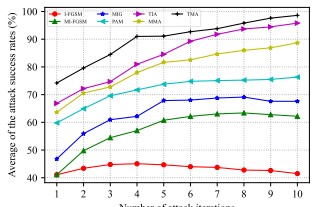

Figure 8: Ablation study of the number of attack iterations on the attack performance. TIA, MMA, and TMA are our proposed attack methods.

### 6.3 ENSEMBLE MODEL ATTACK

There have been many works studying using the ensemble uni-modal model to craft the adversarial example to improve the adversarial transferability (Dong et al., 2018). Here, we conduct experiments to utilize the ensemble of multi-modal, *i.e.*, the audio-visual models, to boost the performance. We iteratively select each one of the 8 models as the victim black-box model and use the remaining 7 models as surrogate models to craft the adversarial examples. We use the adversarial attack success rate against the victim model to evaluate the performance.

As shown in fig. 6, we can notice that the performances of all selected methods are significantly boosted by the ensemble strategy, even surpassing the performance under the white-box attacks. It indicates that *different audio-visual models also share similar areas of interest*. By considering as many surrogate models as possible in attacks, we can fool the target victim model with a high success rate. The previous result shows that the audio-visual model with concat operation as the fusion layer is relatively robust against different attacks. Using the ensemble attacks, the FGSM has an average attack success rate of up to $79.9\%$ against these models, indicating the dimming performance of the concat fusion layer on defense. It should be noted that all our proposed methods achieve an attack success rate of $100\%$ on all victim models, sufficiently demonstrating the effectiveness.

### 6.4 ATTACKING DEFENSE MODELS

Our proposed attack methods, including TIA, MMA, and TMA, have achieved the best attack performance on eight normally trained audio-visual models with different backbones and fusion layers. Recent studies on audio-visual learning proposed to mitigate the threat of audio-visual adversarial examples. In our work, we also propose a bag of novel tricks to enhance adversarial training for defense. Here, to validate the effectiveness of these defenses, as well as show the power of our proposed attack method against the defense mechanism, we conduct white-box adversarial attacks on these eight audio-visual models. For defense methods, we select adversarial training (AT) (Madry et al., 2018), DCFL (Tian & Xu, 2021), Mixup (Li et al., 2022b), CRMT-AT (Yang et al., 2023b), and

the combination of our tricks (Ours). We use our strongest attack TMA to evaluate the robustness. To align the attack setting, we use 10-step PGD adversarial training as the baseline.

The results are shown in fig. 7. Among the different defense methods, training the models on the adversarial examples is more efficient. We can see the adversarial training-based methods, including AT, CRMT-AT, and ours, perform much better than non-adversarial training methods with a clear gap of 10.9%. By integrating our proposed techniques, including the efficient adversarial perturbation crafting and curriculum adversarial training, our adversarial training can further boost the robustness by 2.28% on average versus the runner-up method CRMT-AT.

## 6.5 ABLATION STUDY

**On the number of iterations for the attack**. In experiments, we find that the multistep FGSM, *i.e.*, I-FGSM, cannot always beat FGSM under both white- and black-box settings. This raises the question of whether the number of iterations impacts the adversarial transferability in audio-visual learning. This motivates us to do the ablation study of the number of iterations in attacks.[1] As shown in fig. 8, by generating the adversarial examples on the audio-visual model (AcV), we can see that the number of iterations greatly impacts the attack performance. With increasing the number of iterations, the adversarial transferability of I-FGSM is degraded. While the use of momentum (MI-FGSM) can alleviate the overfitting to the surrogate model, a sufficiently large number of iterations still leads to getting stuck in a local optimum, degrading the performance. The use of input transformations (PAM) can improve the input diversity for better capturing the robust feature, thus boosting the adversarial transferability, but still limited. Our proposed temporal invariant and modality misalignment attack methods sufficiently help the optimization jump over the local optima, thus significantly boosting the performance.

**On the sampling ratio for the adversarial training**. In our approaches, we propose to sample a ratio of frames to generate the temporal-universal adversarial perturbation for efficient adversarial training. There is a balance between the time consumption of adversarial training and the adversarial robustness by selecting different sample ratios. Here, we conduct an ablation study on the sampling ratio. As shown in fig. 9, with increasing the sampling ratio in adversarial training, the defense performance is improved, but it also leads to more time consumption, supporting our argument on the balance. At the same time, a larger sampling ratio does not improve the adversarial robustness much but introduces more time consumption, and a smaller sampling ratio harms defense performance.

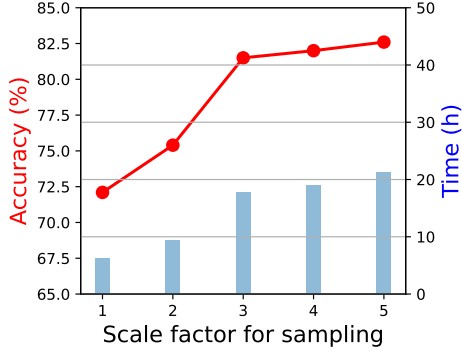

Figure 9: Ablation study of the sampling ratio on the performance of adversarial training. We use our proposed TMA to generate the adversarial example for adversarial training.

## 7 CONCLUSION

In this work, we developed two efficient audio-visual adversarial attack methods: the temporal invariance-based attack and the modality misalignment-based attack. We also introduced an adversarial training framework with strategies for efficient perturbation crafting and curriculum training to reduce time and improve robustness.

Results on the Kinetics-Sounds dataset show that our attacks effectively benchmark audio-visual model robustness and that our training framework improves both robustness and efficiency. Our experiments provided new insights, including the importance of temporal consistency, modality alignment, and the concat fusion layer's robustness. We hope this research will serve as a benchmark for audio-visual robustness and inspire further exploration of AI security in multi-modal data.

---

[1]With the number of iterations as 1, the MI-FGSM and I-FGSM degrade to the FGSM.

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

## A   RESULTS ON MIT-MUSIC DATASET

**Setup**. We use the MIT-MUSIC (Zhao et al., 2018a) for further verification, which contains 685 videos of musical solos and duets. It consists of 11 categories. We split the dataset into 7 : 1 : 2 as the train, validation and test set. Following the same setting in our paper, we train 8 audio-visual models on the MIT-MUSIC dataset and respectively use one surrogate model to attack others under the black-box setting. We use FGSM, I-FGSM, MI-FGSM, MIG, and PAM as our baselines, and compare the attack performance with our proposed methods, including TIA, MMA, and TMA. For defense methods, we select AT, DCFL, Mixup, CRMT-AT, and the baselines.

**Results**. We first use various attack methods to generate adversarial examples by attacking AcR under the white-box setting, and then attack other models under the black-box setting. As shown in fig. 10, our proposed TIA, MMA, and TMA achieve state-of-the-art attack performance among the selected methods. Specifically, TIA achieves an average attack success rate of 95.2%, MMA achieves an average attack success rate of 93.7%, and the combination method TMA achieves an average attack success rate of 97.1%, while that of the runner-up method is 85.1%. We can also see a consistent phenomenon that the adversarial example is easier to transfer between similar architecture, *i.e.*, from AcR to AsR.

We also evaluate the defense performance of our method. The results are depicted in fig. 11. Our proposed method surpasses the baseline methods with a clear margin of 2.2% on defending the adversarial attack versus the runner-up method CRMT-AT.

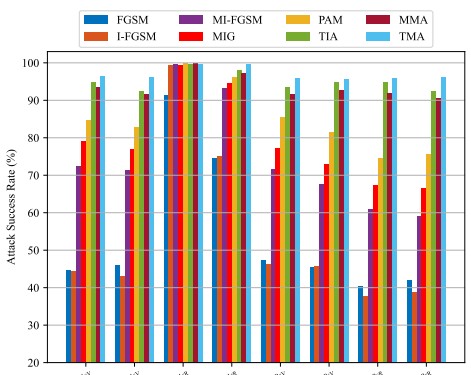

Figure 10: Attack success rates (%) of eight deep models, where the adversarial examples are generated on the white-box surrogate model and attack all models (one white-box model and seven black-box models).

Figure 11: Attack success rates (%) of each of 8 deep models on the adversarial examples crafted under the white-box setting with our proposed TMA method.

## B   ABLATION STUDY ON THE ADVERSARIAL CURRICULUM TRAINING

In our paper, we propose adversarial curriculum training to enhance the performance of adversarial training. This approach includes both data-level and model-level strategies. To study the influence of the scheduler used to adjust the masking ratio (i.e., $\rho_x$ and $\rho_f$), we conduct additional experiments to evaluate the impact of different schedulers on audio-visual robustness. These schedulers are applied to boost the audio-visual adversarial robustness of RsV. Details are as follows:

- **None**: This represents standard adversarial training without employing the adversarial curriculum training strategy.
- **D\*M: 5/20/30%**: These represent constant ratios used in the data- and model-centric strategies. For example:
  - **D\*M: 5%** indicates that 5% of the audio-visual frames and 5% of the model parameters are randomly masked to generate adversarial examples. This setup provides a relatively simple defense scenario for the model.

- **D*M: 20%** means that 20% of the frames and 20% of the parameters are used to generate adversarial examples. This setup creates a stronger adversarial attack and increases the difficulty of adversarial training.

- **D*M: 30%** follows a similar logic with a higher masking ratio for stronger adversarial perturbations.

- **D: 20%**: This indicates that only 20% of the frames are randomly sampled for adversarial example generation during adversarial training, without masking model parameters.

- **M: 20%**: This indicates that 20% of the model parameters are randomly masked in each iteration for adversarial example generation, without randomly dropping frames.

- **M: 40%**: This indicates that 40% of the model parameters are randomly masked in each iteration for adversarial example generation, without randomly dropping frames.

- **Linear**: This applies a linear scheduler from 5% to 20% for both data sampling and model masking ratios.

- **Cosine**: This applies a cosine scheduler from 5% to 20% for both data sampling and model masking ratios.

The results are shown in fig. 12. Compared with vanilla adversarial training, applying the proposed schedulers in adversarial curriculum training enhances adversarial robustness. For instance, when we synchronize masking at 20% on both data and model parameters (D*M: 20%), the success defense rate improves by 3.8%. Furthermore, integrating linear and cosine schedulers to dynamically adjust the difficulty of adversarial examples during training leads to additional improvements of 0.8% and 1.4%, respectively.

It is worth noting that a lower sampling ratio for frames combined with a higher parameter masking ratio can negatively impact the quality of the generated adversarial examples. This, in turn, degrades adversarial robustness, as demonstrated by the results for D*M: 5% and M: 40%.

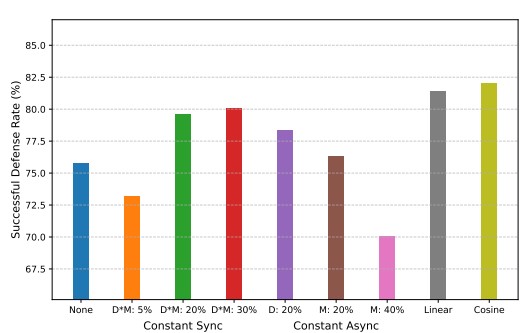

Figure 12: Evaluation results on applying different schedulers to the adversarial curriculum training. The studied model is RsV (ResNet as the visual backbone, VGG as the audio backbone, and sum operation as the fusion layer.)

## C  THE DIFFERENCE BETWEEN VISION-LANGUAGE AND AUDIO-VISUAL ATTACKS/DEFENSE

There are certain similarities between audio-visual attack/defense and vision-language attack/defense methods (Zhang et al., 2022; Lu et al., 2023), as both require consideration of alignment and consistency between the two modalities. However, there are also notable differences between them. (1) Task difference: vision-language attacks aim at attacking content retrieval-related problems while audio-visual attacks focus on classification problems. (2) Operation difference: vision-language attacks perturb input data by optimizing latent embeddings while our method perturbs input by adjusting output logits. (3) Modality difference: vision-language attacks focus on static images while our approach considers the temporal redundancy of dynamic videos. This redundancy motivates our design of curriculum training to exploit sparsity, enhancing adversarial robustness while improving training efficiency.

Considering these reasons, we do not apply our proposed methods to the vision-language domain. However, we believe these works in audio-visual learning are inspiring for other multi-modal areas and we think exploring audio-visual attacks from the perspectives of model pretraining, modality alignment, and content retrieval is a valuable future direction.

# D DEMO APPLICATION ON ATTACKING VIDEOLLAMA

To showcase the scalability of our method in real-world applications, we employ our proposed attack method to generate audio-visual adversarial examples under an ensemble setting. These examples are then used to deceive VideoLLaMA2 (Cheng et al., 2024). Using the prompt, "Please point out the main object generating the sound based on the input video," we evaluate VideoLLaMA's responses. Three results are illustrated in fig. 13, where all inputs are misidentified by VideoLLaMA. For instance, in the case of an airplane, VideoLLaMA incorrectly recognizes it as car racing. This demonstrates the vulnerability of current MLLMs to adversarial attacks, even when the adversarial examples are generated using conventional models.

For a quantitative evaluation, we generated a total of 100 audio-visual adversarial examples using our proposed TMA method and the best baseline method, PAM, respectively, and tested them on VideoLLaMA. While PAM successfully deceived VideoLLaMA in 41 cases, our proposed TMA achieved a higher success rate, with 74 examples successfully attacked.

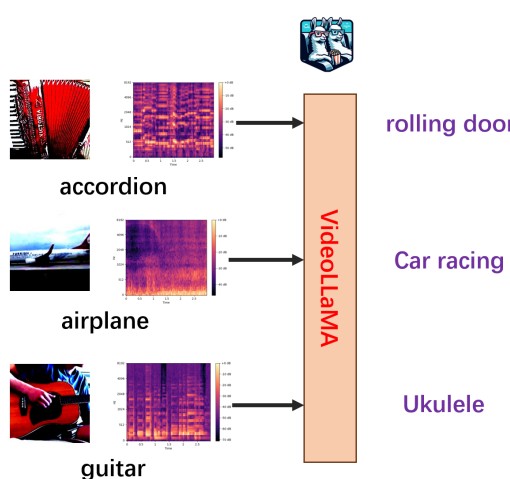

Figure 13: Demo evaluation on attacking the VideoLLaMA2 using the generated audio-visual adversarial examples by our proposed attack method.

