# OpenReview forum: "Rethinking Audio-Visual Adversarial Vulnerability from Temporal and Modality Perspectives"
_ICLR.cc/2025/Conference — ICLR 2025 Poster_

### Official Review · Reviewer_RUCN · 2024-10-22

**Soundness:** 3
**Presentation:** 2
**Contribution:** 3
**Rating:** 6
**Confidence:** 2

**Summary:**

This paper propose two adversarial attack methods for audio-visual data. Audio-visual data has two main characteristics: the temporal consistency (i.e., between nearby time step, the data is similar) and the modality alignment (i.e., audio and visual data has alignment). The attack exploits these two main characteristics: temporal invariance-based attack and modality misalignment-based attack. In addition, this paper propose the defense mechanism that has lower computational cost in comparison to the naive adversarial training. With lower computational cost, the model can be trained with various strength of the attack, improving its robustness. The way to do lower the computational cost is by only attacking sampled frames instead of attacking all frames. Another additional way is to use curriculum learning by masking out the data and the model's weight from low rate (easy) to high rate (hard). The number of steps to generate attacks also go from low steps (easy) to more steps (hard).

**Strengths:**

1. The method of attacking audio-visual data based on the temporal consistency and modality alignment seems to be original.

2. The empirical experiments in Section 3 as motivation is interesting to prove how breaking the temporal consistency and modality alignment (by masking out the data) reduce the audio-visual model performance and increase the attack success rate.

**Weaknesses:**

Several unclear information. Please see Questions.

**Questions:**

1. What is the meaning of (a)synchronous in context of Line 138?

2. Is "contact" in line 411 and 415 is "concat"?

3. Line 366: Is the masking done synchronously or asynchronously?

4. Figure 3. What does it mean minimizing prediction $p_{av}$, $p_a * p_v$ ?

5. Based on Eq. (4), loss in Eq. (2) will be minimized. However, if we want to exploit the temporal consistency (line 52), why not maximizing this Eq. (2) loss? I think, maximizing variation of the extracted features between time step would make the temporal more inconsistent, which can make the attack more efficient.

---

> ### Author Response · Authors · 2024-11-22
>
> Q:
> **[1] What is the meaning of (a)synchronous in context of Line 138?**
> A: In Figs. 1 and 2, we randomly mask frames at a ratio of $\rho$ to evaluate the corruption robustness of audio-visual models. "Synchronous" indicates that we first randomly select frames at the ratio $\rho$  and perturb both the video and audio data together on the selected frames. In contrast, "asynchronous" means perturbing the video and audio on separate segments of the video and audio streams.
>
> **[2] Is "contact" in line 411 and 415 is "concat"?**
> A: Yes. Thanks for your help! We have fixed these typos in our revision.
>
> **[3] Line 366: Is the masking done synchronously or asynchronously?**
> A: The masking is done synchronously. The model is more fragile when the perturbation is synchronous, where the information from different modalities can be complementary under the asynchronous perturbation as shown in Fig. 1 and Fig.2.
>
> **[4] Figure 3. What does it mean minimizing prediction $p_{av}$,$p_a * p_v$?**
> A: Thank you for your question; we understand it might cause some confusion.
>
> - $p_{av}$ represents the model's prediction based on the fusion of audio-visual information. Minimizing $p_{av}$ corresponds to maximizing the classification loss.
> - $p_a \cdot p_v$ measures the alignment between the predictions based solely on audio and visual inputs. Minimizing $p_a \cdot p_v$ aims to generate adversarial perturbations that disrupt the alignment between audio and visual modalities, thereby enhancing the attack's effectiveness.
>
> To make this clearer, we have revised the figure in our paper.
>
>
>
>
> **[5] Based on Eq. (4), loss in Eq. (2) will be minimized. However, if we want to exploit the temporal consistency (line 52), why not maximizing this Eq. (2) loss? I think, maximizing variation of the extracted features between time step would make the temporal more inconsistent, which can make the attack more efficient.**
> A:  The motivation behind minimizing Eq. (2) is to target more robust features along the temporal dimension, which helps improve adversarial transferability.
>
> Maximizing the variation of extracted features between frames is an inspiring idea, as it could diversify the temporal features. However, this approach tends to distract the attack from focusing on robust features, making it easier to fool the surrogate (white-box) model but less effective in attacking victim (black-box) models.
>
> To validate this, we conducted an experiment. Using the same experimental setup, we employed AcA (AlexNet-concat-AlexNet) as the surrogate model to attack RsR (ResNet-sum-ResNet). In a more challenging scenario, we generated adversarial examples using only 5 steps, where our method achieved a white-box attack success rate of 82.3% and a black-box attack success rate of 54.9% against RsR. When we instead maximized Eq. (2) to diversify temporal features, the white-box attack success rate improved to 86.7%, but the black-box attack success rate dropped to 50.5%.
>
> This experiment demonstrates that attacking temporal consistent features is crucial for enhancing audio-visual adversarial transferability.
>
>
> We hope our response addresses your concerns, and we kindly request your consideration in improving the score. Thank you for your valuable review and assistance in improving our revision!

---

### Official Review · Reviewer_nTiz · 2024-11-03

**Soundness:** 3
**Presentation:** 3
**Contribution:** 3
**Rating:** 6
**Confidence:** 4

**Summary:**

The paper presents two powerful adversarial attacks: 1) a temporal invariance attack that exploits the inherent temporal redundancy across consecutive time segments and 2) a modality misalignment attack that introduces incongruence between the audio and visual modalities. To defend against such attacks, the paper introduces a novel audio-visual adversarial training framework. Extensive experiments illustrate the effectiveness of the proposed method.

**Strengths:**

The paper exhibits two powerful adversarial attacks: a temporal invariance attack and a modality misalignment attack. These attacks show their ability to obtain the robustness of audio-visual models.

**Weaknesses:**

The temporal invariance attack is realized by setting consecutive time segments. The modality misalignment attack introduces incongruence between the audio and visual modalities. Regarding the novelty of the two methods, it seems fair. It is encouraging that two attacks achieve SOTA performance.

**Questions:**

1. In experiment comparison, some figures use TMA to represent the proposed method, while some of them use Ours. It is suggested to use the same.
2. For Table 2, does the ablation study use the proposed method TMA? If yes, it is suggested to add the information in the caption.
3. According to Table 2, the successful defense rate continuously increases with the addition of sampling ratio. Where is the change point? The experiment of the ablation study needs to show the sampling ratio value from where the result performance will drop down.

---

> ### Author Response · Authors · 2024-11-22
>
> Q:
> **[1] In experiment comparison, some figures use TMA to represent the proposed method, while some of them use Ours. It is suggested to use the same.**
>
> A: We use "TMA" to represent our proposed attack method and "Ours" to denote our proposed defense method. To minimize confusion, we have clarified this distinction in the figure captions.
>
> **[2] For Table 2, does the ablation study use the proposed method TMA? If yes, it is suggested to add the information in the caption.**
> A: Yes. We have added this information to the caption.
>
> **[3] According to Table 2, the successful defense rate continuously increases with the addition of sampling ratio. Where is the change point? The experiment of the ablation study needs to show the sampling ratio value from where the result performance will drop down.**
>
> A: With increasing the sampling ratio, the successful defense rate will continuously increase until it converges. We propose this strategy based on the temporal redundancy of video and audio modalities, where there are many frames redundant in similar information. While generating the adversarial perturbation for all frames for adversarial training is time-consuming, we only generate the perturbation for a part of them and share the perturbation with neighbor frames to improve the efficiency of adversarial training as well as improve the model robustness. To validate this, we further increase the sampling ratio from 25% to 40%. While the computation overhead increases from 21.3 to 34.2 hours, there is only a 0.4 improvement in the adversarial robustness.
>
>
>
> We hope our response addresses your concerns, and we kindly request your consideration in improving the score. Thank you for your valuable review and assistance in improving our revision!

---

### Official Review · Reviewer_eurp · 2024-11-04

**Soundness:** 3
**Presentation:** 3
**Contribution:** 2
**Rating:** 6
**Confidence:** 4

**Summary:**

This paper introduces two novel audio-visual adversarial attacks, the Temporal Invariance-based Attack (TIA) and the Modality Misalignment-based Attack (MMA), along with an adversarial training framework that enhances robustness and efficiency. The experiments demonstrate the effectiveness of these methods in benchmarking audio-visual model robustness and improving both adversarial robustness and training efficiency.

**Strengths:**

(1) The paper proposes innovative adversarial attacks tailored to audio-visual models, specifically the Temporal Invariance based Attack (TIA) and the Modality Misalignment-based Attack (MMA). These proposed  attacks leverage the unique properties of audio-visual data, such as temporal redundancy and intermodal correlation.

(2) The paper introduces an audio-visual adversarial curriculum training framework that incorporates efficient adversarial perturbation crafting and an adversarial curriculum strategy, which achieves a balance between robustness and efficiency.

**Weaknesses:**

(1) The experiments related to the proposed method are not sufficiently comprehensive.    In Section 5.3, Adversarial Curriculum Training is introduced, which includes two strategies: the Data-level strategy and the Model-level strategy. The core of Adversarial Curriculum Training involves the iterative adjustment of the data masking ratio ( ) and the model dropout ratio ( ). However, the paper does not present corresponding experimental results to substantiate the significance of this iterative design.    Therefore, it is recommended to add experimental results to support the validity of this design.
(2) The paper evaluates unimodal adversarial attack methods (such as FGSM, I-FGSM, and MI-FGSM in the image domain). However, studies such as [1,2] have proposed multimodal attacks. It remains unclear whether the proposed defense method is effective against multimodal adversarial attacks similar to the Audio-Visual attacks discussed in [1].    I suggest adding experiments to evaluate the method's performance under multimodal attack scenarios.

[1] Zhang J, Yi Q, Sang J. Towards adversarial attack on vision-language pre-training models[C]//Proceedings of the 30th ACM International Conference on Multimedia. 2022: 5005-5013.
[2] Lu D, Wang Z, Wang T, et al. Set-level Guidance Attack: Boosting Adversarial Transferability of Vision-Language Pre-training Models[J]. arXiv preprint arXiv:2307.14061, 2023.

**Questions:**

Please refer to the questions raised in the Weakness section. Additionally, the reviewer is interested in the following questions:
(1) What's the selection criterion when choosing datasets for the experiment?
(2) Could the proposed method be incorporated into the pre-trained multi-modal models, such as CLIP or BLIP?

---

> ### Author Response · Authors · 2024-11-22
>
> **W:
> (1) The experiments related to the proposed method are not sufficiently comprehensive. In Section 5.3, Adversarial Curriculum Training is introduced, which includes two strategies: the Data-level strategy and the Model-level strategy. The core of Adversarial Curriculum Training involves the iterative adjustment of the data masking ratio and the model dropout ratio. However, the paper does not present corresponding experimental results to substantiate the significance of this iterative design. Therefore, it is recommended to add experimental results to support the validity of this design.**
>
> A: Thanks for your suggestion! We add this ablation study in Appendix B to verify the significance of our proposed training strategy. Specifically, we evaluate the impacts of different schedulers on audio-visual robustness, including constant and synchronous ratios, constant yet asynchronous asynchronized ratios, a linear scheduler, and a cosine scheduler. Evaluation results demonstrate that by utilizing data-level and model-level strategies and dynamically adjusting the ratios, we effectively boost the robustness of audio-visual models. The reviewer may refer to the appendix B for more details.
>
>
> **(2) The paper evaluates unimodal adversarial attack methods (such as FGSM, I-FGSM, and MI-FGSM in the image domain). However, studies such as [1,2] have proposed multimodal attacks. It remains unclear whether the proposed defense method is effective against multimodal adversarial attacks similar to the Audio-Visual attacks discussed in [1]. I suggest adding experiments to evaluate the method's performance under multimodal attack scenarios.**
>
> A: Thank you for your insightful point! There are certain similarities between audio-visual attack/defense and vision-language attack/defense methods, as both require consideration of alignment and consistency between the two modalities. However, there are also notable differences between them. (1) Task difference: vision-language attacks aim at attacking content retrieval-related problems while audio-visual attacks focus on classification problems. (2) Operation difference: vision-language attacks perturb input data by optimizing latent embeddings while our method perturbs input by adjusting output logits. (3) Modality difference: vision-language attacks focus on static images while our approach considers the temporal redundancy of dynamic videos. This redundancy motivates our design of curriculum training to exploit sparsity, enhancing adversarial robustness while improving training efficiency.
>
> Considering these reasons, we do not apply [1, 2] to the audio-visual domain. However, we believe these works are inspiring and we think exploring audio-visual attacks from the perspectives of model pretraining, vision-language alignment, and content retrieval is a valuable future direction. We have included this discussion in Appendix C. Thank you for the suggestion!
>
>
> [1] Zhang et al. Towards adversarial attack on vision-language pre-training models[C]/ACM MM 2022.
> [2] Lu et al. Set-level Guidance Attack: Boosting Adversarial Transferability of Vision-Language Pre-training Models[J]. arXiv preprint arXiv:2307.14061, 2023.
>
> **Q: Please refer to the questions raised in the Weakness section. Additionally, the reviewer is interested in the following questions: (1) What's the selection criterion when choosing datasets for the experiment? (2) Could the proposed method be incorporated into the pre-trained multi-modal models, such as CLIP or BLIP?**
>
> A: (1) We choose datasets based on their generality and popularity. Both Kinetics-Sounds and MIT-Music datasets are widely used in the audio-visual area [1, 2, 3]. To further verify the scalability of our proposed method, we additionally conduct experiments on the MIT-Music dataset, with results provided in Appendix A.
> [1] Tian et al. "Can audio-visual integration strengthen robustness under multimodal attacks?." CVPR 2021.
> [2] Li et al. "On adversarial robustness of large-scale audio visual learning." ICASSP 2022.
> [3] Yang et al. "Quantifying and enhancing multi-modal robustness with modality preference." ICLR 2024.
>
> (2) Both the proposed attack and defense methods primarily leverage temporal sparsity and redundancy to enhance performance, which is more pronounced in audio-visual data than in vision-language data. Exploring the presence of redundancy in vision-language modalities and developing adversarial training strategies for pre-trained multi-modal models with limited data is an inspiring research direction. While our current work focuses on audio-visual data, we have included a discussion of this perspective in the appendix, with the aim of motivating further research in this area.
>
> We hope our response addresses your concerns, and we kindly request your consideration in improving the score. Thank you for your valuable review and assistance in improving our revision!

---

### Official Review · Reviewer_7ZEx · 2024-11-05

**Soundness:** 3
**Presentation:** 2
**Contribution:** 3
**Rating:** 6
**Confidence:** 3

**Summary:**

This paper studies audio-visual adversarial vulnerability, and proposes two kinds of adversarial attacks which deals with the temporal consistency and intermodal correlation. Furthermore, a new audio-visual adversarial training framework is also designed to defend against above attacks.

**Strengths:**

1. The two kinds of proposed adversarial attacks are novel and important for the topic of audio-visual adversarial vulnerability.
2. The experiments shows the effectiveness of the proposed attacks.

**Weaknesses:**

1. The proposed defending method seems not designed exactly towards the proposed attacks. In particular, the modality-misalignment-based attack is not specifically addressed.

**Questions:**

Maybe it is better to show the effectiveness of the method by some real audio-visual samples, i.e., applying the proposed attacks can indeed attack existing methods while the proposed defending method can handle/alleviate it.

---

> ### Author Response · Authors · 2024-11-22
>
> ***W:* The proposed defending method seems not designed exactly towards the proposed attacks. In particular, the modality-misalignment-based attack is not specifically addressed.**
>
> A:  During the adversarial training process, we utilize our proposed Temporal and Modality-based Attack (TMA), which includes modality-misalignment-based attack, to craft adversarial examples and train the model on these generated examples. As a result, the trained model can effectively defend against modality-misalignment-based attacks.
>
> The primary challenge in audio-visual adversarial training lies in efficiency. Therefore, our focus has been on proposing a new strategy to enhance efficiency by leveraging temporal redundancy. We understand this may have caused some confusion, and we have revised the paper to clarify this point.
>
>
> ***Q:* Maybe it is better to show the effectiveness of the method by some real audio-visual samples, i.e., applying the proposed attacks can indeed attack existing methods while the proposed defending method can handle/alleviate it**
>
> A: Thank you for your suggestion. To demonstrate the scalability of our method in real-world applications, we utilized our proposed attack method, TMA, to generate 100 audio-visual adversarial examples to deceive the multi-modal large language model, VideoLLaMA2 [1], achieving an attack success rate of 74%. Detailed experimental results are provided in Appendix D.
>
> In our work, we designed an efficient adversarial training method that leverages audio-visual redundancy to enhance adversarial robustness while improving training efficiency. However, due to limited computational resources and restricted access to the pre-training datasets of current MLLMs, it is challenging to apply adversarial training directly to foundation models like VideoLLaMA2. We hope this research inspires future studies to explore more efficient approaches for improving adversarial robustness from both temporal and modality perspectives.
>
>
> We hope our response addresses your concerns, and we kindly request your consideration in improving the score. Thank you for your valuable review and assistance in improving our revision!
>
> [1] Cheng, Zesen, et al. "VideoLLaMA 2: Advancing Spatial-Temporal Modeling and Audio Understanding in Video-LLMs." arXiv preprint arXiv:2406.07476 (2024).

---

> > ### Comment · Reviewer_7ZEx · 2024-11-29
> >
> > Thanks for the detailed response, which addresses most of my concerns. Taking into account the response and the reviews by other reviewers, I maintain my rating.

---

> > > ### Author Response · Authors · 2024-11-29
> > >
> > > Thank you for your valuable feedback and for taking the time to review our paper. We are glad to hear that we have successfully addressed most of your concerns：）
> > >
> > > Please feel free to let us know if there are any remaining questions or concerns about our paper.

---

### Author Response · Authors · 2024-11-22

**To all reviewers:**
We sincerely thank all reviewers for their valuable feedback and for recognizing the merits of our work:
1. We propose novel methods for audio-visual adversarial attack/defense, leveraging the unique characteristics of audio-visual learning (7ZEx, eurp, RUCN).
2. The evaluation experiments demonstrate strong consistency, and the results validate the effectiveness of our approach (7ZEx, nTiz, RUCN).

In our revision, we have made the following updates:

- **Expanded Application on attacking the MLLM**: We include a demonstration of our proposed Temporal and Modality-based Attack (TMA) on the multi-modal large language model, VideoLLaMA2, achieving a high attack success rate. Details are provided in Appendix D.
- **Enhanced Discussion**: We add a discussion on the relationships between various multi-modal adversarial attacks in Appendix C, aiming to inspire further research on robustness in multi-modal domains.
- **Additional Ablation Studies**: We include more ablation studies in Table 2 and Appendix B.
- **Improved Clarity**: We revise both illustration figures and their captions for better clarity.
- **Typos Corrected**: We fix all identified typographical errors.

We hope these updates address your concerns and further strengthen the contributions of our work. Thank you again for your thoughtful reviews and support!

---

### Author Response · Authors · 2024-11-26
**Kind reminder for discussion**

Dear Reviewers,

We sincerely appreciate your time and effort in reviewing our paper and providing valuable suggestions, which have greatly helped us improve our work.

As the deadline for the paper revision period approaches, we wanted to check if we have adequately addressed your concerns and made necessary revisions to our paper. If you have any remaining questions or feedback, please feel free to share them with us. We are deeply grateful for this opportunity to exchange ideas with you and greatly value your insights, which have been instrumental in enhancing our work.

Thank you once again for your time and thoughtful feedback!

Best regards,
Authors

---

### Meta-Review · Area_Chair_4xrM · 2024-12-17

**Metareview:**

The paper is above the threshold for acceptance due to its novel contributions to audio-visual adversarial vulnerability, including the Temporal Invariance Attack (TIA) and Modality Misalignment Attack (MMA), which leverage unique properties of audio-visual models and achieve SOTA performance. The proposed adversarial curriculum training framework, while needing more comprehensive evaluation, introduces innovative strategies for balancing robustness and efficiency. Strong experimental results and impactful insights into model vulnerabilities outweigh minor concerns, making this work a significant contribution to the field.

**Additional Comments On Reviewer Discussion:**

N/A

---

### Decision · Program_Chairs · 2025-01-22

Accept (Poster)